# Prognostic Impact of Atrial Fibrillation in Patients with Heavily Calcified Coronary Artery Disease Receiving Rotational Atherectomy

**DOI:** 10.3390/medicina59101808

**Published:** 2023-10-11

**Authors:** Jin Jung, Yeonjoo Seo, Sung-Ho Her, Jae-Hwan Lee, Kyusup Lee, Ki-Dong Yoo, Keon-Woong Moon, Donggyu Moon, Su-Nam Lee, Won-Young Jang, Ik-Jun Choi, Jang-Hoon Lee, Sang-Rok Lee, Seung-Whan Lee, Kyeong-Ho Yun, Hyun-Jong Lee

**Affiliations:** 1Department of Cardiology, St. Vincent’s Hospital, College of Medicine, The Catholic University of Korea, Seoul 16247, Republic of Korea; colaking@naver.com (J.J.); yookd@catholic.ac.kr (K.-D.Y.); cardiomoon@gmail.com (K.-W.M.); babaheesu@gmail.com (D.M.); yellow-night@hanmail.net (S.-N.L.); raph83@naver.com (W.-Y.J.); 2Department of Internal Medicine, Yeouido St. Mary’s Hospital, College of Medicine, The Catholic University of Korea, Seoul 07345, Republic of Korea; yj.seo.md@gmail.com; 3Department of Cardiology in Internal Medicine, Chungnam National University School of Medicine, Chungnam National University Sejong Hospital, Sejong 30099, Republic of Korea; 4Department of Cardiology, Daejeon St. Mary’s Hospital, College of Medicine, The Catholic University of Korea, Seoul 34943, Republic of Korea; ajobijh@hanmail.net; 5Department of Cardiology, Incheon St. Mary’s Hospital, College of Medicine, The Catholic University of Korea, Incheon 21431, Republic of Korea; mrfasthand@catholic.ac.kr; 6Department of Internal Medicine, Kyungpook National University Hospital, Daegu 41944, Republic of Korea; ljhmh75@knu.ac.kr; 7Department of Cardiology, Chonbuk National University Hospital, Jeonju 54907, Republic of Korea; medorche@naver.com; 8Department of Cardiology, Asan Medical Center, University of Ulsan College of Medicine, Seoul 05505, Republic of Korea; seungwlee@amc.seoul.kr; 9Department of Cardiovascular Medicine, Regional Cardiocerebrovascular Center, Wonkwang University Hospital, Iksan 54538, Republic of Korea; ards7210@wonkwang.ac.kr; 10Department of Internal Medicine, Sejong General Hospital, Bucheon 14754, Republic of Korea; untouchables00@hanmail.net

**Keywords:** percutaneous coronary intervention, rotational atherectomy, atrial fibrillation, clinical outcomes, coronary artery disease

## Abstract

*Background and Objectives*: Although both rotational atherectomy (RA) and atrial fibrillation (AF) have a high thrombotic risk, there have been no previous studies on the prognostic impact of AF in patients who undergo percutaneous coronary intervention (PCI) using RA. Thus, the aim of the present study was to determine the prognostic impact of AF in patients undergoing PCI using RA. *Materials and Methods*: A total of 540 patients who received PCI using RA were enrolled between January 2010 and October 2019. Patients were divided into AF and sinus rhythm groups according to the presence of AF. The primary endpoint was net adverse clinical events (NACEs) defined as a composite outcome of all-cause death, myocardial infarction, target vessel revascularization, cerebrovascular accident, or total bleeding. *Results*: Although in-hospital adverse events showed no difference between those with AF and those without AF (in-hospital events, 54 (11.0%) vs. 6 (12.2%), *p* = 0.791), AF was strongly associated with an increased risk of NACE at 3 years (NACE: hazard ratio, 1.880; 95% confidence interval, 1.096–3.227; *p* = 0.022). *Conclusions*: AF in patients who underwent PCI using RA was strongly associated with poor clinical outcomes. Thus, more attention should be paid to thrombotic and bleeding risks.

## 1. Introduction

Atrial fibrillation (AF) is the most common arrhythmia. Global prevalence of AF continues to increase with population aging [1]. AF is a well-known risk factor for athero-thrombotic adverse vascular events, and oral anticoagulant is recommended for preventing these events in patients with AF [2]. AF is also considered a major public health challenge because of its strong association with cardiovascular morbidity and mortality [1,3,4,5].

As with AF, the prevalence of coronary artery disease (CAD) is increasing [6]. Because risk factors for AF and CAD overlap, patients with AF are more likely to have comorbid CAD than patients with sinus rhythm (SR) control [7]. AF is also a marker of advanced coronary atherosclerosis. Patients with AF are often treated with percutaneous coronary intervention (PCI) [4,8]. Since patients undergoing PCI with AF require concurrent antiplatelet and anticoagulation treatment and coexistence of AF and CAD worsens the prognosis, the management of these patients can be challenging [9].

Thus, various previous studies have reported the optimal antithrombotic strategy for AF patients who undergo PCI [10,11]. In addition, previous studies of associations between AF and clinical outcomes after PCI have been conducted in various patient groups, including unstable angina [12], acute coronary syndrome (ACS) [13,14], ST segment elevation myocardial infarction (STEMI) [15], and all presentations for PCI [16,17].

However, there have been no previous studies on the prognostic impact of AF in patients who undergo PCI using rotational atherectomy (RA) due to severe coronary artery calcification (CAC). Although both RA and AF have a high thrombotic risk, the prognostic impact of AF in patients receiving RA is unclear. Thus, the aim of the present study was to determine the prognostic impact of AF in patients undergoing PCI using RA.

## 2. Materials and Methods

The study population consisted of 540 patients with heavily calcified CAD who underwent PCI using RA from January 2010 to October 2019 at 9 medical centers in Korea. Patients were included within the ROtational atherectomy in Calcified lesion in Korea (ROCK) registry, which was approved by the Institutional Review Board (IRB) of each hospital. Data were collected at each medical center using a standardized case report form to record baseline characteristics and procedure-related and follow-up data. Follow-up data were collected up to 3 years retrospectively based on patients’ medical records and/or telephone interviews.

Patients were divided into AF and SR groups according to the presence of AF. The flow chart is displayed in Figure 1. Baseline characteristics and clinical outcomes were compared between the two groups. AF was defined as the documented AF on an electrocardiogram (ECG) during the hospitalization, previously diagnosed AF, or AF confirmed via ECG in the medical record. Electrocardiographic AF was defined as having no defined P-waves and the presence of an irregular rhythm with fibrillatory wave. The primary endpoint was net adverse clinical events (NACEs) defined as a composite outcome of all-cause death, myocardial infarction (MI), target vessel revascularization (TVR), cerebrovascular accident (CVA), or total bleeding. Secondary endpoints were all-cause death, cardiac death, MI, any repeat revascularization (RR), TVR, CVA, and total bleeding. RR was defined as any surgical or percutaneous revascularization in any vessel. TVR was defined as any surgical or percutaneous revascularization of the treated vessel. CVA was defined as a focal neurological deficit of central origin lasting >24 h, confirmed by imaging and a neurologist. Chronic kidney disease (CKD) was defined as an estimated glomerular filtration rate < 60 mL/min/1.73 m^2^, as calculated using the Modification of Renal Diet equation from baseline serum creatinine. 

Procedural outcomes and in-hospital events including death, urgent coronary artery bypass grafting (CABG)/PCI, periprocedural MI, or CVA during the hospitalization period were also examined. All definitions were the same as previously published in the ROCK registry study [18].

All RA procedures were performed using a RotablatorTM RA system (Boston Scientific, Marlborough, MA, USA) guided by standard techniques. During RA, a short duration of burr rotation runs and intracoronary nitroglycerin and/or verapamil were used to prevent slow/no reflow phenomenon and coronary spasm. Treatment strategies related to procedural details including selection of vascular access, decisions regarding timing of RA, burr number, and burr size were dependent on the discretion of attending operators. Patients’ management before and after PCI was performed in accordance with accepted guidelines and standard care [19]. All clinical events were confirmed by source documentation collected at each enrolled medical center and centrally adjudicated by an independent group of clinicians unaware of the revascularization type.

Continuous variables are presented as median and interquartile range or mean ± standard deviation and were analyzed using Student’s *t*-test or the Mann–Whitney test. Categorical variables are expressed as numbers and percentages. Differences between two groups were compared using Fisher’s exact test or chi-square test. Univariable/multivariable Cox regression analyses were performed. For multivariate analysis, confounding factors were age, sex, CKD, left ventricle ejection fraction (LVEF), and new oral anticoagulant (NOAC). Hazard ratio (HR) and 95% confidence interval (CI) were also calculated. Event rates were estimated using Kaplan–Meier estimates in time-to-first-event analyses and compared using the log-rank test. A *p*-value < 0.05 was considered statistically significant. All statistical analyses were performed using Statistical Analysis Software (SAS) version 9.2 (SAS Institute, Cary, NC, USA).

## 3. Results

### 3.1. Baseline Characteristics

Baseline characteristics of AF and SR groups are presented in Table 1 and Table 2. There was no significant difference in baseline demographic and clinical characteristics between the two groups except for age, CKD, and LVEF. Compared with patients without AF, patients with a history of AF were more likely to be older and to have CKD, lower LVEF (age, 71.1 ± 10.2 vs. 75.2 ± 8.6, *p* = 0.006; CKD, 82 (16.7%) vs. 14 (28.6%), *p* = 0.038; LVEF, 53.4 ± 13.1 vs. 48.7 ± 15.5, *p* = 0.018). There was no difference between the two groups in baseline angiographic characteristics except for arc of calcification > 270° (left main disease, 68 (13.9%) vs. 6 (12.2%), *p* = 0.756; MVD, 385 (78.4%) vs. 39 (79.6%), *p* = 0.848; IVUS, 220 (44.8%) vs. 29 (59.2%), *p* = 0.054; and procedure time, 79.5 ± 51.1 vs. 76.8 ± 45.9, *p* = 0.725). Arc of calcification > 270 degrees evaluated intravascular ultrasound (IVUS) was 56.9% in the SR group and 81.8% in the AF group (arc of calcification > 270°, 83/143 (56.9%) vs. 18/22 (81.8%), *p* = 0.026). In angiography, severe CAC was defined as radiopacity noted without cardiac motion before contrast dye injection compromising both sides of the coronary arterial lumen, which is known to be equivalent to about 215 degrees of arc of calcification in IVUS evaluation [20]. Therefore, it can be seen that most patients in both groups had severe calcified coronary artery disease (CAD) on angiographic grading.

### 3.2. In-Hospital Events and Procedural Outcomes

In-hospital events and procedural outcomes are also presented in Table 3. There were no differences in in-hospital events between the two groups (in-hospital events, 54 (11.0%) vs. 6 (12.2%), *p* = 0.791; in-hospital death, 10 (2.0%) vs. 1 (2.0%), *p* > 0.999; urgent CABG/PCI, 9 (1.8%) vs. 0 (0.0%), *p* > 0.999; in-hospital CVA, 1 (0.2%) vs. 1 (2.0%), *p* = 0.173). In particular, periprocedural MI occurred more frequently in the AF group, but there was no statistical significance (periprocedural MI, 39 (7.9%) vs. 6 (12.2%), *p* = 0.281). Procedural outcomes also showed no difference between the two groups except for the temporal pacemaker (tPM) insertion (coronary dissection, 69 (14.1%) vs. 9 (18.4%), *p* = 0.413; coronary perforation, 10 (2.0%) vs. 0 (0.0%), *p* = 0.611; procedure success, 472 (96.1%) vs. 48 (98.0%), *p* > 0.999: tPM, 11 (2.2%) vs. 5 (10.2%), *p* = 0.010). In-hospital bleeding occurred more frequently in AF, but there was no statistical difference (in-hospital bleeding, 23 (4.7%) vs. 4 (8.2%), *p* = 0.294).

### 3.3. Clinical Outcomes

Compared to patients without AF, NACE, the primary endpoint, occurred more frequently in patients with AF (NACE, 100 (20.4%) vs. 21 (42.9%), *p* < 0.001). Any RR and TVR also occurred nearly twice as often in the AF group (any RR, 46 (9.4%) vs. 8 (16.3%), *p* = 0.062; TVR, 38 (7.7%) vs. 7 (14.3%), *p* = 0.055). In the univariable analysis, any RR and TVR together with NACE occurred significantly more in the AF group compared to the SR group (any RR: hazard ratio (HR) 2.268, 95% confidence interval (CI) 1.061–4.849, *p* = 0.018; TVR: HR 2.268, 95% CI 1.061–4.849, *p* = 0.035). After multivariable adjustment for confounding factors including age, sex, CKD, LVEF, and NOAC, the presence of AF was still associated with NACE, unlike RR and TVR, showing a 1.880-fold risk compared to patients without AF (NACE: HR 1.880, 95% CI 1.096–3.227, *p* = 0.022; any RR: HR 1.971, 95% CI 0.843–4.609, *p* = 0.118; TVR: HR 2.259, 95% CI 0.891–5.728, *p* = 0.086). On the other hand, total bleeding occurred almost twice as often in AF, but no statistical significance was shown (total bleeding, 26 (5.3%) vs. 6 (12.2%), *p* = 0.059) (Table 4) (Figure 2).

## 4. Discussion

The principal findings of the present study were as follows: (1) among patients who underwent PCI using RA due to severe CAC, 9.07% (49/540) had AF; (2) in-hospital adverse events and complications including bleeding showed no difference between those with AF and those without AF; (3) however, AF was strongly associated with an increased risk of NACE at 3 years. 

Asian countries are known to have a lower incidence of AF than Western countries [2,21]. However, 9.07% of patients had AF in this study. This percentage was similar to or higher than those reported for Western countries [16,17,22]. This was also higher than those reported for East Asian countries [23,24]. This result might be because patients in this study had severe CAC requiring RA, and the more severe the CAC, the higher the AF incidence rate [25]. Although the association between CAC and AF has not been fully elucidated yet, patients with severe CAC have enlarged left atrium and pulmonary veins, suggesting an association of severe CAC with AF [26].

As the number of elderly patients increases, patients with severe CAC, as well as those with AF, will also increase [27]. Severe CAC may impair device delivery including balloon, stent, and IVUS, and also damage the polymer drug coating, resulting in impaired drug delivery [28]. In particular, severe CAC impedes stent expansion, and stent under-expansion was found to be one of the leading causes of in-stent restenosis in a previous study, as evaluated by IVUS [29]. For these reasons, patients with severe CAC showed poor post-PCI outcomes compared to those with non-calcified lesions [30]. In these patients, RA is a useful option for procedural success via the modification of heavily calcified plaque to facilitate pre-balloon and stent delivery and achieve sufficient stent expansion [27,31].

However, the complexity of the procedure and concerns about complications make it difficult to use RA. This is because thrombi and micro-debris occurring after RA can induce microembolization and slow/no reflow and increase the risk of thrombotic events [32,33]. In particular, AF patients are at greater risk than SR patients. This is because the study targets patients with severe CAC, which requires RA, but AF patients generally have higher CAC scores [34], associated with complications such as bleeding and periprocedural MI [35,36], compared to SR patients. The present study also showed the same trend. The proportion of patients with arc of calcification > 270° according to IVUS was higher in AF patients. In addition, since patients with AF also have a high thrombogenic risk, causing no reflow more often than in patients with SR [37], if patients who need RA also have AF, concerns about complications in PCI using RA will increase. However, in the present study, there was no statistically significant difference in in-hospital events or complications, except for tPM insertion between the two groups. In other words, even AF patients should actively consider PCI using RA to overcome heavily calcified lesions without worrying about complications. However, microembolization of small plaques and calcified particles that occur during PCI with RA might lead to bradycardia and conduction abnormalities. Nonetheless, it was not recommended to routinely perform tPM insertion in contemporary RA due to the lack of proven clinical benefit [27]. However, the right coronary artery procedure, associated with ischemia from sinoatrial nodal or atrioventricular nodal arteries, had a significantly higher risk of a conduction abnormality, so prophylactic pacemaker insertion was considered according to the operator’s decision [38,39]. According to the results of our study, it is better to consider tPM insertion in advance for a safe procedure in AF patients, not only for procedures on the right coronary artery.

Unlike in-hospital events and complications, the incidence of NACE was significantly higher in the AF group than in the SR group. Consistent with previous studies [16,17,23,24], our results showed that AF patients were older, had a higher prevalence of CKD and lower LVEF, and AF was associated with poor clinical outcomes. Relatively short-term (1 year) and long-term (6 year) studies [23,24] compared to this study also showed the same results. Takashi Miura et al. [24], like in this study, conducted a study in East Asia and showed poorer prognosis with more thrombotic and bleeding events in AF patients compared to patients without AF. In particular, as in our study, multivariate analysis showed that AF was strongly associated with an increased risk of NACE at 1 year. Choi et al. [23], in a study conducted in Korea, demonstrated that patients with AF who underwent PCI had poorer long-term outcomes. As a result, this indicated that adverse effects of AF were maintained regardless of the time period. There were several possible explanations for the poorer clinical outcome in patients with AF. AF can cause hemodynamic impairments including increased O_2_ consumption, microvascular dysfunction, loss of atrial contraction, and increased left atrial filling pressure [40,41,42]. Irregular RR intervals can promote atrioventricular block (which is also a well-known complication of RA) and malignant arrhythmia [13]. AF is also known to increase systemic inflammation [41]. Increased inflammation is associated with the development of CAC by inflammation-related cytokines, including interleukin-6, interleukin-8, and interleukin-13 [43]. Increased inflammation is also strongly associated with the risk of coronary events [44,45]. In our study, coronary events, including TVR, RR, and MI, showed a tendency to occur more frequently in the AF group. A recent study has demonstrated that high-sensitivity C-reactive protein, a representative inflammation marker, is a stronger predictor for risk of future cardiovascular events and death in patients taking statins than cholesterol assessed by LDL cholesterol, which is already well known to have a strong correlation with cardiovascular disease [44]. As the importance of inflammation increased, the CANTOS trial [46] was published, showing that coronary artery events can be reduced through anti-inflammatory treatment. Therefore, inflammation management to reduce coronary events will also be important in AF patients.

Contrary to previous studies [16,17,23,24], our study did not show a statistically significant difference in bleeding events between the two groups. This might be because the number of patients in the AF group was relatively small. Thus, there was no statistical difference in bleeding events between the two groups, although bleeding events occurred twice in the AF group. In addition, while NOAC was used as an anticoagulant in this study, other studies [17,23] used vitamin K antagonist, which might have also led to different results. To reduce bleeding events in AF patients who have undergone PCI, future research on antithrombotic treatment of optimal intensity and duration is needed.

## 5. Limitation

This study had several limitations. First, as this was an observational and retrospective study, our results might be vulnerable to selection bias. Second, to determine the presence of AF, we reviewed medical records and clinical examinations, including the Holter and electrocardiogram tests. In this way, AF prevalence might have been underestimated because there may be patients with paroxysmal AF that were not confirmed in the examination. In addition, this study did not subclass AF into new onset or not, or paroxysmal or persistent AF, which could affect results [13]. Third, medications were investigated at discharge, but specific antiplatelet and antithrombotic strategies, such as dual antiplatelet treatment, single antiplatelet treatment, and NOAC, or triple treatment, were not investigated after discharge. Finally, the proportion of subjects in the AF group was relatively small. Therefore, caution is required when interpreting our results.

## 6. Conclusions

This study firstly demonstrated that among patients receiving PCI and using RA due to heavily calcified lesions, AF was not rare and was associated with poor clinical outcomes. In these patients, RA is a useful device for a successful procedure. However, more attention should be paid to thrombotic and bleeding risks. More research is needed to reduce these risks.

## Figures and Tables

**Figure 1 medicina-59-01808-f001:**
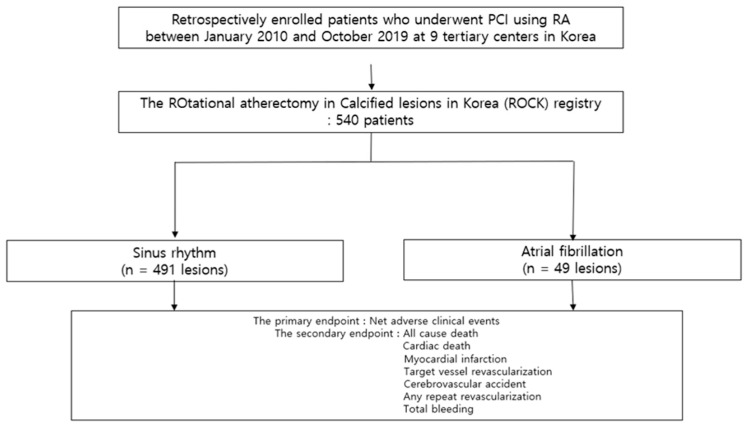
Study population flow chart. PCI, percutaneous coronary intervention; RA, rotational atherectomy.

**Figure 2 medicina-59-01808-f002:**
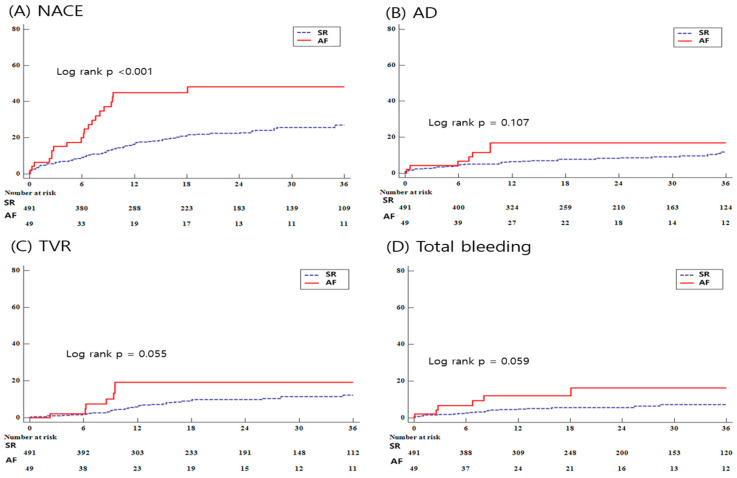
Kaplan–Meier curve for clinical outcomes during follow up. NACE, net adverse clinical event; AD, all-cause death; TVR, target vessel revascularization.

**Table 1 medicina-59-01808-t001:** Baseline demographic and clinical characteristics.

	SR (*n* = 491)	AF (*n* = 49)	*p*-Value
Age, years	71.1 ± 10.2	75.2 ± 8.6	0.006
Sex			0.925
Male	294 (59.9)	29 (59.2)	
Female	197 (40.1)	20 (40.8)	
BMI	24.2 ± 3.9	24.3 ± 4.0	0.952
Smoking	96 (19.6)	7 (14.3)	0.371
HTN	375 (76.4)	40 (81.6)	0.405
DM	278 (56.6)	27 (55.1)	0.838
Hyperlipidemia	218 (44.4)	17 (34.7)	0.191
CKD	82 (16.7)	14 (28.6)	0.038
Dialysis	44 (9.0)	5 (10.2)	0.793
Previous PCI	126 (25.7)	13 (26.5)	0.895
Previous CABG	22 (4.5)	2 (4.1)	>0.999
Previous MI	61 (12.4)	5 (10.2)	0.651
CVA	64 (13.0)	11 (22.5)	0.069
Clinical diagnosis			0.510
Stable angina	297 (60.5)	32 (65.3)	
ACS	194 (39.5)	17 (34.7)	
LVEF, %	53.4 ± 13.1	48.7 ± 15.5	0.018
NOAC	2 (0.4)	14 (28.6)	<0.001
DAPT	473 (96.3)	46 (93.9)	0.426
Beta blocker	347 (70.7)	33 (67.4)	0.627
ACEi/ARB	316 (64.4)	25 (51.0)	0.065
Statin	457 (93.1)	45 (91.8)	0.767
HbA1C	6.7 ± 1.4	6.7 ± 1.6	0.973
Total cholesterol	144.0 ± 39.1	140.2 ± 34.3	0.538
LDL cholesterol	84.8 ± 40.3	84.0 ± 29.8	0.897
HDL cholesterol	46.0 ± 14.3	47.0 ± 16.5	0.660
Triglyceride	119.7 ± 75.8	119.7 ± 54.9	1.000

Data are shown as mean ± SD or *n* (%). SR, sinus rhythm; AF, atrial fibrillation; BMI, body mass index; HTN, hypertension; DM, diabetes mellitus; CKD, chronic kidney disease; PCI, percutaneous coronary intervention; CABG, coronary artery bypass graft; MI, myocardial infarction; CVA, cerebrovascular accident; ACS, acute coronary syndrome; LVEF, left ventricle ejection fraction: NOAC, new oral anticoagulant; DAPT, dual antiplatelet treatment; ACEi/ARB, angiotensin-converting enzyme inhibitors/angiotensin II receptor blockers; HbA1c, glycated hemoglobin.

**Table 2 medicina-59-01808-t002:** Baseline angiographic characteristics.

	SR (*n* = 491)	AF (*n* = 49)	*p*-Value
ACC/AHA classification			0.228
A	2 (0.4)	1 (2.0)	
B1	39 (7.9)	1 (2.0)	
B2	47 (9.6)	5 (10.2)	
C	403 (82.1)	42 (85.7)	
Left main disease	68 (13.9)	6 (12.2)	0.756
MVD	385 (78.4)	39 (79.6)	0.848
IVUS	220 (44.8)	29 (59.2)	0.054
Arc of calcification > 270°	83/143 (56.9)	18/22 (81.8)	0.026
Procedure time, min	79.5 ± 51.1	76.8 ± 45.9	0.725

Data are shown as mean ± SD or *n* (%). SR, sinus rhythm; AF, atrial fibrillation; ACC/AHA, American College of Cardiology/American Heart Association; MVD, multi-vessel disease; IVUS, intravascular ultrasound.

**Table 3 medicina-59-01808-t003:** In-hospital events and procedural outcomes.

	SR (*n* = 491)	AF (*n* = 49)	*p*-Value
In-hospital events	54 (11.0)	6 (12.2)	0.791
In-hospital death	10 (2.0)	1 (2.0)	>0.999
Urgent CABG/PCI	9 (1.8)	0 (0.0)	>0.999
Periprocedural MI	39 (7.9)	6 (12.2)	0.281
In-hospital CVA	1 (0.2)	1 (2.0)	0.173
Procedural outcomes			
Coronary dissection *	69 (14.1)	9 (18.4)	0.413
Temporary pacemaker during procedure	11 (2.2)	5 (10.2)	0.010
Coronary perforation	10 (2.0)	0 (0.0)	0.611
In-hospital bleeding	23 (4.7)	4 (8.2)	0.294
Procedure success	472 (96.1)	48 (98.0)	>0.999

Data are shown as mean ± SD or *n* (%). * Coronary dissection from defined from The National Heart, Lung, and Blood Institute (NHLBI) classification system. SR, sinus rhythm; AF, atrial fibrillation; CABG, coronary artery bypass grafting; PCI, percutaneous coronary intervention; MI, myocardial infarction; CVA, cerebrovascular accident.

**Table 4 medicina-59-01808-t004:** Clinical outcomes and univariable/multivariable cox regression analysis.

	SR (*n* = 491)	AF (*n* = 49)	*p*-Value	Univariate HR (95% CI)	*p*-Value	Multivariate HR ** (95% CI)	*p*-Value
NACE	100 (20.4)	21 (42.9)	<0.001	2.219 (1.404–3.507)	<0.001	1.880 (1.096–3.227)	0.022
All-cause death	37 (7.9)	7 (14.3)	0.107	1.636 (0.737–3.633)	0.226	1.396 (0.584–3.339)	0.453
Cardiac death	28 (5.7)	2 (4.1)	0.712	0.627 (0.150–2.612)	0.521	0.645 (0.150–2.784)	0.557
Myocardial infarction	17 (3.5)	3 (6.1)	0.273	1.779 (0.524–6.042)	0.356	1.335 (0.316–5.644)	0.694
Any repeat revascularization	46 (9.4)	8 (16.3)	0.062	2.268 (1.151–4.470)	0.018	1.971 (0.843–4.609)	0.118
TVR	38 (7.7)	7 (14.3)	0.055	2.268 (1.061–4.849)	0.035	2.259 (0.891–5.728)	0.086
CVA	8 (1.6)	2 (4.1)	0.228	2.273 (0.491–10.521)	0.294	0.320 (0.012–8.291)	0.492
Total bleeding	26 (5.3)	6 (12.2)	0.059	2.125 (0.884–5.108)	0.092	1.950 (0.774–5.111)	0.175

** adjusted by age, sex, chronic kidney disease (CKD), left ventricle ejection fraction (LVEF), and new oral anticoagulant (NOAC). Data are shown as mean ± SD or *n* (%). SR, sinus rhythm; AF, atrial fibrillation; HR, hazard ratio; CI, confidence interval, NACE, net adverse clinical events; TVR, target vessel revascularization; CVA, cerebrovascular accident.

## Data Availability

Not applicable.

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
