# Peer review of "Prognostic Impact of Atrial Fibrillation in Patients with Heavily Calcified Coronary Artery Disease Receiving Rotational Atherectomy"

_medicina, 2023, doi:10.3390/medicina59101808_

Round 1
Reviewer 1 Report
A very interesting study, well designed and performed.
Please explain the NOAC/OAC treatment in the AF group - who only less than 30% of pmts had NOAC at baseline? was AF diagnosed during or after the procedure? what was the pre and postpocedural NOAC / OAC and SAPT or DAPT regimen?
What were the reasons for NOAC in non-AF group ?
Author Response
Point 1. Please explain the NOAC/OAC treatment in the AF group - who only less than 30% of pmts had NOAC at baseline? was AF diagnosed during or after the procedure? what was the pre and postpocedural NOAC / OAC and SAPT or DAPT regimen?
Response 1:. Thank you for your valuable comment. In baseline characteristics, medication is based on the medication prescribed at the time of patient discharge. At discharge, DAPT was prescribed in 94% of patients and NOAC was prescribed in 27.8% of patients in the AF group. The reason why the number of patients who were prescribed NOAC despite the presence of AF was low was because most of the patients were prescribed DAPT, probably because they were concerned about the risk of bleeding due to triple treatment. However, after discharge, the patients' antiplatelet and antithrombotic strategies were not analyzed. As you pointed out, this is also a limitation of our study. I will add the contents of this to the limitation.
As mentioned in the method section, the AF group includes both patients who were previously diagnosed with AF and patients who were confirmed to have AF during hospitalization, regardless of whether before or after the procedure.
“AF was defined as the documented AF on an electrocardiogram (ECG) during the hos-pitalization, previously diagnosed AF, or AF confirmed on ECG in the medical record.”
Modified text
Third, medications were investigated at discharge, but specific antiplatelet and antithrombotic strategies, such as dual antiplatelet treatment , single antiplatelet treamtent and NOAC, or triple treatment, were not investigated after discharge.
Point 2. What were the reasons for NOAC in non-AF group ?
Response 2:. Both cases had remanent thrombus-like lesions in final angiography, so NOAC was used despite the absence of AF.

Reviewer 2 Report
Line 42 “95% confidence interval, 1.054–2.968; p = 0.31).” p=0.31?
Line 42. “Conclusions: AF was not rare and…” Very strange conclusion, how rare/not rare?
Line73 “heavily calcified” – how it was measured? It is focus of the study, worth paying attention.
Line 99 “Continuous variables were presented as median and interquartile range or mean ± standard deviation using Student’s t-test.” What do you mean?
How many patients with AF were included in the study?
As we understand number of lesions was bigger than number of patients as some patients had several lesion. 529 lesions in the SR group….. but how many males and females in reality? Does it mean that in table 1 patients with several lesion are included several times when for example mean BMI is calculated etc? Please include numbers of patients.
The same question is for results in other tables and figure 2. Adverse effect like myocardial infarction or bleeding take place in patient or in lesion? How many lesions patients with AF had? Did they had more lesions than patients with SR? Did you analyze if the frequency of NACE was dependent on number of lesions in AF patients? Did they had non heavily calcified lesions which were not subjected to RA? How they might influence adverse events in the future?
Line 170 “Compared to patients without AF, NACE, the primary endpoint, occurred more frequently in patients with AF (112 [21.2%] vs. 22 [40.7%], p < 0.001).” Here we are talking about patients or lesions?
Line 189 “Principal findings of the present study were as follows: 1) among patients who underwent PCI using RA due to severe CAC, 10.2% of patients had AF” where is this information in Results? 10.2 % of 540 patients it is 55. How it can be 54 lesions in AF group, additionally as some patients had several lesions?
Author Response
Response to Reviewer Comments
Point 1. Line 42 “95% confidence interval, 1.054–2.968; p = 0.31).” p=0.31?
Response 1: We confirmed that it was entered incorrectly and corrected it to the correct number.
Original text
AF was strongly associated with an increased risk of NACE at 3 years (hazard ratio, 1.769; 95% confidence interval, 1.054–2.968; p = 0.31).
Modified text
AF was strongly associated with an increased risk of NACE at 3 years (hazard ratio, 1.880; 95% confidence interval, 1.096–3.227; p = 0.022)
Point 2. Line 42. “Conclusions: AF was not rare and…” Very strange conclusion, how rare/not rare?
Response 2. As mentioned in the discussion, AF has a higher prevalence in western compared to Asia. However, the AF rate in the subjects in this study was similar to that in the subjects in the western study. This appears to be due to the reasons mentioned in the text. "This result might be because patients in this study had severe CAC requiring RA and the more severe the CAC, the higher the AF incidence rate [25]. "
Therefore, this phrase was inserted to emphasize this.
However, as you commented, it seems somewhat inappropriate to be added to abstract. The content will be deleted from the abstract.
Point 3. Line73 “heavily calcified” – how it was measured? It is focus of the study, worth paying attention.
Response 3:. I agree with your point. In other RA-related studies, angiographic grading “radiopacities noted without cardiac motion before contrast injection generally compromising both sides of the arterial lumen” is mainly used as the definition of severe or heavy calcification. In real clinical practice (Korea), according to insurance standards, RA is used only when 1) IVUS does not pass or the balloon does not pass through the lesion, 2) when the balloon does not expand more than 2mm. So most of the calcified vessels using rotational atherectomy may satisfy this definition and there is no dispute that these vessels are severely or heavily calcified. And we have some data that evaluated whether the arc of calcium was more than 270 degrees with IVUS, and we will added it to the new table for baseline angiographic characteristics. Severe calcification in angiography is comparable 215 degree on IVUS evaluation [1]. Taking this into account, it can be confirmed that much more patients have severe calcification in angiography grading.
[1] Mintz GS, Popma JJ, Pichard AD, Kent KM, Satler LF, Chuang YC, et al. Patterns of calcification in coronary artery disease. A statistical analysis of intravascular ultrasound and coronary angiography in 1155 lesions. Circulation. 1995; 91: 1959–1965
Modified text
In angiography, severe CAC was defined as radiopacity noted without cardiac motion before contrast dye injection compromising both sides of the coronary arterial lumen, which is known to be equivalent to about 215 degrees of arc of calcification in IVUS evaluation [20]. Therefore, it can be seen that most patients in both groups had severe calcified coronary artery disease (CAD) on angiographic grading.
Point 4.Line 99 “Continuous variables were presented as median and interquartile range or mean ± standard deviation using Student’s t-test.” What do you mean?
Response 4: The phrase has been modified to clarify the meaning
Original text
Continuous variables were presented as median and interquartile range or mean ± standard deviation using Student’s t-test
Modified text
Continuous variables were presented as median and interquartile range or mean ± standard deviation and analyzed using Student’s t-test or Mann-Whitney test.
Point 5.
How many patients with AF were included in the study?
As we understand number of lesions was bigger than number of patients as some patients had several lesion. 529 lesions in the SR group.. but how many males and females in reality? Does it mean that in table 1 patients with several lesion are included several times when for example mean BMI is calculated etc? Please include numbers of patients.
The same question is for results in other tables and figure 2. Adverse effect like myocardial infarction or bleeding take place in patient or in lesion? How many lesions patients with AF had? Did they had more lesions than patients with SR? Did you analyze if the frequency of NACE was dependent on number of lesions in AF patients? Did they had non heavily calcified lesions which were not subjected to RA? How they might influence adverse events in the future?
Line 170 “Compared to patients without AF, NACE, the primary endpoint, occurred more frequently in patients with AF (112 [21.2%] vs. 22 [40.7%], p < 0.001).” Here we are talking about patients or lesions?
Line 189 “Principal findings of the present study were as follows: 1) among patients who underwent PCI using RA due to severe CAC, 10.2% of patients had AF” where is this information in Results? 10.2 % of 540 patients it is 55. How it can be 54 lesions in AF group, additionally as some patients had several lesions?
Response 5: All of these questions appear to have arisen because the present analysis is confusing whether it is lesion-specific or patients-specific. Several articles on the ROCK registry have been published, and most of the articles have performed statistical analysis based on lesion, and this article is also the same.
However, after reviewing your comments, it seems that the article is causing a confusion because it was analyzed as a lesion. To clarify the content of the article, the data were reanalyzed based on the patients. Among patients with AF, there were few patients with multiple lesions, so there was no change in the major results even when statistical analysis was performed based on patient

Round 2
Reviewer 2 Report
Authors reanalyzed the data and answered all questions so the manuscript can be accepted.